# Positioning Combination Method of USBL Using Four-Element Stereo Array

**DOI:** 10.3390/s21227722

**Published:** 2021-11-20

**Authors:** Wei Wang, Min Zhu, Bo Yang

**Affiliations:** 1Ocean Acoustic Technology Laboratory, Institute of Acoustics, Chinese Academy of Sciences, Beijing 100190, China; wangwei1988@mail.ioa.ac.cn (W.W.); yangbo@mail.ioa.ac.cn (B.Y.); 2Beijing Engineering Technology Research Center of Ocean Acoustic Equipment, Beijing 100190, China; 3University of Chinese Academy of Sciences, Beijing 100049, China; 4State Key Laboratory of Acoustics, Institute of Acoustics, Chinese Academy of Sciences, Beijing 100190, China; 5Major Mission Department, Institute of Acoustics, Chinese Academy of Sciences, Beijing 100190, China

**Keywords:** ultra-short baseline (USBL), four-element stereo array, cross-planar array, combined location method, positioning performance

## Abstract

In the present article, an ultra-short baseline (USBL) combined location method based on three four-element stereo arrays is proposed. In order to solve the problem of the poor positioning effect of acoustic positioning under a high incident angle of signal, two kinds of four-element stereo arrays are designed, and the localization approach of the new array is given. At the same time, for the regular triangular pyramid array, a virtual array element is proposed to construct a planar cross array to improve the poor positioning effect of the regular triangular pyramid array at a low incident angle. This paper analyzes the positioning performance of three arrays. Combined with the traditional cross-planar array localization method, a set of positioning strategies to switch the two localization methods under different incident angles were designed. The switching thresholds of the two methods were analyzed by simulation. Simulation results show that the new arrays can locate stably at different incident angles and improve the overall positioning performance of the array.

## 1. Introduction

With the development of marine technology, the demand for deep-sea operation is increasing. Underwater acoustic localization plays a more and more important role in underwater operation. Because of its small size, low cost, strong portability and simple operation, the ultra-short baseline positioning system has been widely used in the fields of marine exploration and scientific research, underwater operation and marine resources development [1,2,3,4,5,6].

The traditional ultra-short baseline localization method uses a four-element planar array to estimate the location of the target by measuring the phase difference or time delay difference within channels and the slant ranging from the target to the array [7,8]. Zheng Cuie [9] proposed a method by using the phase anti ambiguity of double-pulse signal to improve the positioning accuracy, which reduces the number of array elements while increasing the complexity of the transmitter at the same time. Zheng Enming [10] optimized the traditional array elements, basically using small spacing array elements to solve ambiguity and large spacing array elements to improve positioning accuracy. Luo Qinghua [11,12] introduced Kalman filter-based methodology on the array designed by Zheng Enming to improve the positioning accuracy, but these improvements do not improve the poor localization performance of signals at a large incident angle. Liang Guolong [13] proposed using depth information to assist localization to improve the positioning accuracy by up to 0.25%. This method requires additional depth information, and there is no accuracy improvement in some angles. The localization performance of the four-element planar array is poor or even unable to locate the signal when the signal has a high incident angle. Therefore, researchers presented the regular triangular pyramid array [14,15,16,17]. Although the regular triangular pyramid array solves the problem of poor localization performance at a large incident angle, it also reduces the localization performance at a small incident angle. Zhang Xu [18] designed a four-element stereo array that calculated the depth in a similar way to the horizontal in the vertical direction, which improved the localization performance of the signal at a high incident angle and reduced the localization performance of the signal at a low incident angle. 

These algorithms of the planar array have poor positioning accuracy in the vertical direction when the signal has a high incident angle, resulting in poor overall positioning accuracy. The algorithms of the stereo array improve the positioning accuracy under a high incident angle, but there is a problem of poor positioning accuracy when the signal has a low incident angle. To solve the abovementioned problems two new arrays are proposed in this paper to improve the localization performance when the signal has a high incident angle. At the same time, based on the regular triangular pyramid array, a four-element plane array is virtualized to improving the poor localization effect at a low incident angle. In summary, the main contributions of this paper are as follows:(1)In view of the poor positioning accuracy of the plane array under a high incident angle, two new arrays were designed, and a vector projection algorithm based on the new array is given. The appropriate formation parameters are found through simulation.(2)Aiming at the problem of poor positioning accuracy of regular triangular pyramid array at a low incident angle, a virtual quaternion plane array algorithm is proposed.(3)The three arrays adopt the combination algorithm of vector projection method and cross-array method, which can not only solve the localization problem of signal at a high incident angle, but also ensure that the positioning accuracy of a signal at a low incident angle is not reduced.

## 2. Three Arrays and Algorithm Principle 

In order to solve the problem of poor localization performance of signal at a high incident angle, it is necessary to change the traditional planar array into stereo array. The traditional four-element planar array is shown in Figure 1, in which the spacing of array elements 1, 2, 3 and 4 is d. In this paper, two methods are used to change the planar array into a stereo array. One method is to fold the plane and keep ∠BAD and ∠BCD at right angles. The included angle between plane BAD and plane BCD is 2α. The new array is shown in Figure 2.

Here, the distance between array elements 2 and 4 is $\sqrt{2}d$. The depth of array elements 1 and 3 is always 0. The angle between edge 12 and edge 14 remains unchanged at 90°. The coordinates of the four elements are (0, $-\frac{d}{\sqrt{2}}\sin\alpha$, 0), ($\frac{d}{\sqrt{2}}$, 0, $\frac{d}{\sqrt{2}}cos\alpha$), (0, $\frac{d}{\sqrt{2}}\sin\alpha$, 0), ($-\frac{d}{\sqrt{2}}$, 0, $\frac{d}{\sqrt{2}}cos\alpha$). The center of the array is considered to be at the coordinate origin. The red dot shows the location of the source. The distance between the source and the center of the array is r. The azimuth of the incident signal is φ, and the pitch is θ. The coordinates of the source can be expressed as (rsinθcosφ, rsinθsinφ, rcosφ). The unit vector of source direction is (sinθcosφ, sinθsinφ, cosθ). The vectors composed of array elements $l_{12}=(\frac{d}{\sqrt{2}},\frac{d}{\sqrt{2}}\sin\alpha ,\frac{d}{\sqrt{2}}\cos\alpha )$, $l_{13}=(0,\frac{d}{\sqrt{2}}2\sin\alpha ,0)$, $l_{14}=(-\frac{d}{\sqrt{2}},\frac{d}{\sqrt{2}}\sin\alpha ,\frac{d}{\sqrt{2}}\cos\alpha )$. By calculating the projection of each vector in the source direction, the relationship between the incident azimuth and pitch angle of the signal and the reception delay can be obtained. The details are as follows:(1)$$d_{12}=S\cdot l_{12}=\frac{d}{\sqrt{2}}\sin\theta \cos\varphi +\frac{d}{\sqrt{2}}\sin\alpha \sin\theta \sin\varphi +\frac{d}{\sqrt{2}}\cos\alpha \cos\theta =\left(\tau_{1}-\tau_{2}\right)\times c$$
(2)$$d_{13}=S\cdot l_{13}=\frac{d}{\sqrt{2}}2\sin\alpha \sin\theta \sin\varphi =\left(\tau_{1}-\tau_{3}\right)\times c$$
(3)$$d_{14}=S\cdot l_{14}=-\frac{d}{\sqrt{2}}\sin\theta \cos\varphi +\frac{d}{\sqrt{2}}\sin\alpha \sin\theta \sin\varphi +\frac{d}{\sqrt{2}}\cos\alpha \cos\theta =\left(\tau_{1}-\tau_{4}\right)\times c$$
(4)$$d_{42}=d_{12}-d_{14}=\sqrt{2}d\sin\theta \cos\varphi =\left(\tau_{4}-\tau_{2}\right)\times c$$
where $d_{12}$, $d_{13}$, $d_{14}$, $d_{42}$ represent the projection of vectors $l_{12}$, $l_{13}$, $l_{14}$, and $l_{42}$ in the signal direction, respectively, which can be obtained by multiplying the delay difference of each channel by the sound velocity. Use Formula (1) + Formula (3) to obtain:(5)$$d_{12}+d_{14}=\sqrt{2}d\sin\alpha \sin\theta \sin\varphi +\sqrt{2}d\cos\alpha \cos\theta$$

Substitute Formula (2) into Formula (5) to obtain:(6)$$\theta =\mathrm{acos}(\frac{d_{12}+d_{14}-d_{13}}{\sqrt{2}d\cos\alpha })$$

Formula (2) is divided by Formula (4) to obtain:(7)$$\varphi =\mathrm{atan}(\frac{d_{13}}{d_{42}\sin\alpha })$$

Then the coordinates of the source under the carrier coordinates can be obtained:(8)x=rsinθcosφ
(9)y=rsinθsinφ
(10)z=rcosφ

According to Formula (6), under the influence of noise when $\frac{d_{12}+d_{14}-d_{13}}{\sqrt{2}d\cos\alpha }$ is more than 1, the pitch θ is invalid, thus localization failed at this time. This occurs when the incident angle of the signal is close to 0°. When the incident angle of the signal θ is high, this method can obtain effective θ and φ.

Another way to change the array is to keep the distance between array elements 1,3 and array elements 2,4 unchanged. At the same time, the height of array elements 2 and 4 is down to obtain a stereo array. The second new array is shown in Figure 3.

The distance between array elements 1,3 and 2,4 is $\sqrt{2}d$, and the included angle between plane BAD and plane BCD is 2α. The coordinates of the four elements are (0, $-\frac{d}{\sqrt{2}}$, 0), ($\frac{d}{\sqrt{2}}$, 0, $\frac{d}{\sqrt{2}\tan\alpha }$), (0, $\frac{d}{\sqrt{2}}$, 0), ($-\frac{d}{\sqrt{2}}$, 0, $\frac{d}{\sqrt{2}\tan\alpha }$). The vectors composed of array elements $l_{12}=(\frac{d}{\sqrt{2}},\frac{d}{\sqrt{2}},\frac{d}{\sqrt{2}\tan\alpha })$, $l_{13}=(0,\sqrt{2}d,0)$, $l_{14}=(-\frac{d}{\sqrt{2}},\frac{d}{\sqrt{2}},\frac{d}{\sqrt{2}\tan\alpha })$. Here, the red dot indicates the location of the source. Similarly, the relationship between the incident azimuth and pitch angle of the signal and the reception delay can be obtained by vector projection. The details are as follows:(11)$$d_{12}=S\cdot l_{12}=\frac{d}{\sqrt{2}}\sin\theta \cos\varphi +\frac{d}{\sqrt{2}}\sin\theta \sin\varphi +\frac{d}{\sqrt{2}\tan\alpha }\cos\theta =\left(\tau_{1}-\tau_{2}\right)\times c$$
(12)$$d_{13}=S\cdot l_{13}=\frac{d}{\sqrt{2}}2\sin\theta \sin\varphi =\left(\tau_{1}-\tau_{3}\right)\times c$$
(13)$$d_{14}=S\cdot l_{14}=-\frac{d}{\sqrt{2}}\sin\theta \cos\varphi +\frac{d}{\sqrt{2}}\sin\theta \sin\varphi +\frac{d}{\sqrt{2}\tan\alpha }\cos\theta =\left(\tau_{1}-\tau_{4}\right)\times c$$
(14)$$d_{42}=d_{12}-d_{14}=\sqrt{2}d\sin\theta \cos\varphi =\left(\tau_{4}-\tau_{2}\right)\times c$$

Formula (11) + Formula (13) – Formula (12) can obtain:(15)$$\theta =a\cos(\frac{d_{12}+d_{14}-d_{13}}{\sqrt{2}d}\tan\alpha )$$

Divide Equation (12) by Equation (14) to obtain:(16)$$\varphi =a\tan(\frac{d_{13}}{d_{42}})$$

Similarly, the source coordinates can be calculated according to formulas (8)–(10). The general regular triangular pyramid array is shown in the Figure 4:

The side length of the tetrahedron is d, and the element 4 is located above the origin. The coordinates of the four elements are($\frac{\sqrt{3}}{3}d$,0, 0), ($-\frac{\sqrt{3}}{6}d$, $-\frac{d}{2}$, 0), ($-\frac{\sqrt{3}}{6}d$, $\frac{d}{2}$, 0), (0, 0, $-\frac{\sqrt{6}}{3}d$). The coordinates of the source can be expressed as (rsinθcosφ, rsinθsinφ, rcosφ). The vectors composed of array elements $l_{12}=(-\frac{\sqrt{3}}{2}d,-\frac{1}{2}d,0)$, $l_{13}=(-\frac{\sqrt{3}}{2}d,\frac{1}{2}d,0)$, $l_{14}=(-\frac{\sqrt{3}}{3}d,0,-\frac{\sqrt{6}}{3}d)$.
(17)$$d_{12}=S\cdot l_{12}=-\frac{\sqrt{3}}{2}d\sin\theta \cos\varphi -\frac{1}{2}d\sin\theta \sin\varphi =\left(\tau_{1}-\tau_{2}\right)\times c$$
(18)$$d_{13}=S\cdot l_{13}=-\frac{\sqrt{3}}{2}d\sin\theta \cos\varphi +\frac{1}{2}d\sin\theta \sin\varphi =\left(\tau_{1}-\tau_{3}\right)\times c$$
(19)$$d_{14}=S\cdot l_{14}=-\frac{\sqrt{3}}{3}d\sin\theta \cos\varphi -\frac{\sqrt{6}}{3}d\cos\varphi =\left(\tau_{1}-\tau_{4}\right)\times c$$

According to Equations (17)–(19):(20)$$\theta =a\cos(\frac{-3d_{14}+d_{12}+d_{13}}{\sqrt{6}d})$$
(21)$$\varphi =a\tan(\frac{d_{12}-d_{13}}{d_{13}+d_{12}}\sqrt{3})$$

The azimuth estimation formula of the triangular pyramid array is similar to that of stereo arrays 1 and 2, which can improve the signal localization performance at a high incident angle but also has the problem of poor localization performance at a low incident angle. In order to improve this problem, a virtual array element will be introduced. The specific structure is shown in the Figure 5:

The green dot is a virtual element, which is obtained by the symmetry of the element 1 under the connection between elements 2 and 3. The projection of the vector $l_{15}$ in the signal direction is:(22)$$d_{15}=S\cdot l_{15}=\left(\tau_{1}-\tau_{5}\right)\times c$$

In addition, $l_{15}=l_{12}+l_{13}$, then:(23)$$d_{15}=S\cdot l_{15}=S\cdot \left(l_{12}+l_{13}\right)=S\cdot l_{12}+S\cdot l_{13}=\left(\tau_{1}-\tau_{2}\right)\times c+\left(\tau_{1}-\tau_{3}\right)\times c$$

According to the parallel wave algorithm:(24)$$x=-\frac{\left(\tau_{1}-\tau_{2}+\tau_{1}-\tau_{3}\right)\times c}{\sqrt{3}d}\times r-\frac{\sqrt{3}}{6}d$$
(25)$$y=-\frac{\left(\tau_{3}-\tau_{2}\right)\times c}{d}\times r$$
(26)$$z=\sqrt{r^{2}-x^{2}-y^{2}}$$

## 3. Simulation Analysis of Algorithm Performance

Because the localization performance of two stereo arrays will be different under different included angles 2α, it is necessary to simulate and analyze the specific localization effect of the stereo array under different included angles. During the simulation, the array element spacing d was 0.32 m and the source was the LFM signal with a center frequency of 10 kHz and bandwidth of 5 kHz. The sampling rate of the received signal was 80 kHz. The specific processing flow of the algorithm is shown in the Figure 6:

Considering that the vector projection method is mainly used when the signal has a high incident angle, and the parallel wave method is mainly used when the signal has a low incident angle, the coordinate points (400, −400, 10) and (10, −10,100), respectively, were selected for the source location during the simulation, and the signal-to-noise ratio was 20 dB. Each included angle was simulated 50 times. The (400, −400, 10) location adopted the vector projection method, and the (10, −10,100) location adopted the parallel wave method.

As can be seen from Figure 7, in the process of the half-included angle of the stereo array 1 from low to high, the positioning accuracy of the signal at a high incident angle first gradually improves and then decreases, and the positioning accuracy of the signal at a low incident angle gradually improves. The overall localization performance of stereo array 1 is better when the half-included angle is about 60°.

As can be seen from Figure 8, similar to stereo array 1, in the process of the half-included angle of the stereo array 2 from low to high, the positioning accuracy of the signal at a high incident angle first gradually improves and then decreases, and the positioning accuracy of the signal at a low incident angle gradually improves. The overall localization performance of the stereo array 2 is better when the half-included angle is about 70°. According to the above simulation results, stereo arrays 1 and stereo array 2 have better performance when the half-included angle is 60° and 70°, respectively. Later, we will compare the localization performance of the 60° half-included angle stereo array 1 and the 70° half-included angle stereo array 2 with the traditional cross array of the corresponding size.

The simulation area is set to 400 × 400 m, the grid size is 5 × 5 m, the depth is 15 m, the signal-to-noise ratio is 20 dB, and the number of simulations per location is 50. The results are shown in the Figure 9, Figure 10 and Figure 11:

The stereo array 3 in the Figure 11 refers to the triangular pyramid array. Comparing the positioning accuracy of the parallel wave method and vector projection method of the above three kinds of stereo arrays, it can be seen that the positioning accuracy of the parallel wave method decreases gradually when the signal incident angle increases from small to large and becomes worse when the incident angle is large; the positioning accuracy of the vector projection method decreases gradually when the signal incident angle changes from high to low and becomes worse when the incident angle is low. In order to further analyze the localization performance of the two algorithms of the three arrays at different incident angles, the signal source was selected at x = 100 m, y = −200 m and different depth positions for the simulation.

It can be seen from the Figure 12 that when the signal incidence angle is 0–70°, the positioning accuracy of the cross array parallel wave method is better; at 80–90°, the stereo array 1 vector projection method is better.

It can be seen from Figure 13 that when the incident angle is 0–80°, the positioning accuracy of the cross array parallel wave method is better; At 80–90°, the positioning accuracy of the stereo array 1 vector projection method is better.

It can be seen from the Figure 14 that when the incident angle is 0–40°, the positioning accuracy of the cross array parallel wave method is better; the positioning accuracy of stereo array 3 vector projection method is better at 40–90°. Generally speaking, the performance of the vector projection method is better than that of the cross array at a high incident angle, and that of the cross array method is better than that of the vector projection method at a low incident angle. In practice, the combination of stereo array vector projection method and cross-array algorithm can be considered.

## 4. Simulation of Combined Localization Performance

Combining the localization performance of the vector projection method and parallel wave method at different incident angles, the vector projection method is used when the incident angle is high, and the parallel wave method is used when the incident angle is low. For stereo array 1, when the incident angle is greater than 80°, the algorithm switches from the parallel wave method to the vector projection method. If the incident angle of the signal decreases from high to low, and when the incident angle is less than 80°, the algorithm switches from the vector projection method to the parallel wave method. For stereo array 2, if the signal incident angle increases from small to large when the incident angle is greater than 83°, the algorithm switches from the parallel wave method to the vector projection method. If the incident angle of the signal decreases from high to low when the incident angle is less than 83°, the algorithm switches from the vector projection method to the parallel wave method. For stereo array 3, if the incident angle of the signal increases from low to high when the incident angle is greater than 40°, the algorithm switches from the parallel wave method to the vector projection method. If the incident angle of the signal decreases from high to low when the incident angle is less than 40°, the algorithm switches from the vector projection method to the parallel wave method. The simulation range is 400 × 400 m, the grid size is 5 × 5 m, the depth is 15 m, and the number of single-point simulations is 50. The results of the combined algorithm and the algorithm in literature 18 are shown in the Figure 15, Figure 16, Figure 17 and Figure 18:

Comparing Figure 9 and Figure 15a, Figure 10 and Figure 16a, Figure 11 and Figure 17, it can be seen that compared with the original algorithm, the combination algorithm of the three stereo arrays improves the positioning accuracy of the vector projection method when the signal has a low incident angle and improves the positioning accuracy of the parallel wave method when the signal has a high incident angle. Comparing Figure 15a,b and Figure 16a,b, the combined algorithm of stereo array 1 and stereo array 2 has better positioning accuracy than the algorithm in literature 18 when the incident angle is low. When the signal has a high incident angle, the positioning accuracy is almost the same as that of the algorithm in literature 18. Comparing Figure 15a, Figure 16a and Figure 18, it can be seen that compared with the planar array algorithm, the combination algorithm of the stereo arrays improves the positioning accuracy when the signal has a high incident angle. In order to further analyze the localization performance of the two algorithms of the three arrays at different incident angles, the signal source was selected at x = 100 m, y = −200 m and different depth positions for simulation and compared with the stereo array parallel wave method in literature 18. Because the triangular pyramid array is not suitable for the algorithm in literature 18, the triangular pyramid array was only compared before and after the algorithm combination.

As can be seen from Figure 19 and Figure 20, the combined algorithm of stereo arrays 1 and 2 has better localization performance than the algorithm in literature 18. When the incident angle is less than 70°, the positioning accuracy of the combined algorithm of stereo array 1 is 0.02–0.04% higher than that of literature 18. When the incident angle is greater than 80°, the positioning accuracy of the two is close. When the incident angle is less than 85°, the positioning accuracy of the combined algorithm of stereo array 2 is 0.03–0.2% higher than that of literature 18. When the signal incidence angle is greater than 85°, the positioning accuracy of the two is close. Figure 21 shows that compared with the precombination algorithm, the positioning accuracy of the stereo array 3 combination algorithm is improved by 0.35% at a small incidence angle and 1.12% at a large incidence angle. Figure 22 compares the positioning accuracy of the combination algorithm of three stereo arrays. The positioning accuracy of stereo array 2 is highest at a low incident angle and that of stereo array 3 is highest at a high incidence angle. Overall, the positioning accuracy of stereo array 3 is uniform, all below 0.06%.

The localization efficiency in terms of localization time was evaluated and compared with other related methods, i.e., the algorithm in literature 18, the orthogonal 8-element array, the non-equidistant quaternary array and the orthogonal quaternary array. The localization processing time is illustrated in Table 1.

It can be seen from Table 1 that the localization processing time of the non-equidistant quaternary array is the shortest, followed by the orthogonal quaternary array followed. The orthogonal quaternary array adopts dual-frequency pulses, and the amount of calculation is a little more than that of non-equidistant quaternary array, because the number of elements of the orthogonal 8-element array is twice that of a four-element array, and the localization processing time is about twice that of the four-element array. The localization processing time of the four stereo arrays are almost the same. For the four-element stereo array, due to the use of broadband signal processing, it needs to calculate the correlation results between the received signal and the local signal, which requires more time. From the simulation results, the localization processing time of the four-element stereo array is about twice that of four-element planar array.

## 5. Conclusions

Because the traditional cross array cannot locate or has poor localization performance at a high incident angle, two stereo arrays’ positioning algorithms are proposed in this paper. The localization performance of the two stereo arrays under different half-included angles was analyzed, and the appropriate half-included angle was selected to compare the performance with the traditional cross array. Aiming at the poor localization performance of the regular triangular pyramid array at a low incident angle, a positioning method of a virtual four primitive plane array is proposed by using vector synthesis. For the above three arrays, the combination algorithm of the vector projection method and parallel wave method was used to improve the localization performance as a whole, and the parameters of algorithm switching were given by simulation. The simulation results show that compared with the orthogonal 8-element array, the non-equidistant quaternary array and orthogonal quaternary array, stereo arrays 1, 2 and 3 improve the localization accuracy at a high incident angle. Compared with the stereo array algorithm in literature 18, the combination algorithm of stereo arrays 1 and 2 have higher localization accuracy at a low incident angle. The stereo array 3 combination algorithm improves the localization accuracy of the original algorithm at a low incident angle. Compared with the combination algorithms of stereo array 1, 2 and 3, the localization accuracy of stereo array 3 is generally uniform, all of which are below 0.06%. It can be concluded that the proposed combination method of stereo array outperforms the other USBL methods.

In the following research, we will implement the USBL positioning system with the combination algorithms proposed in this paper. We will develop the software and hardware platform of the USBL system and test it in pools and lakes.

## Figures and Tables

**Figure 1 sensors-21-07722-f001:**
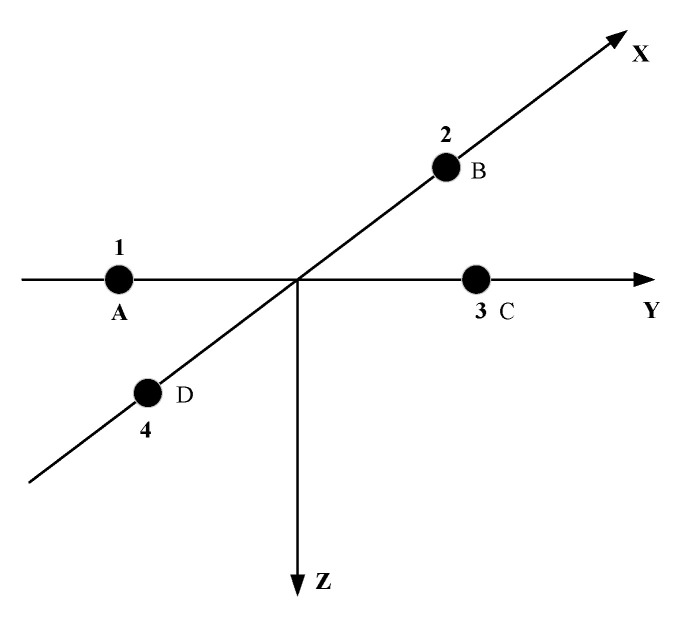
Traditional four-element planar array.

**Figure 2 sensors-21-07722-f002:**
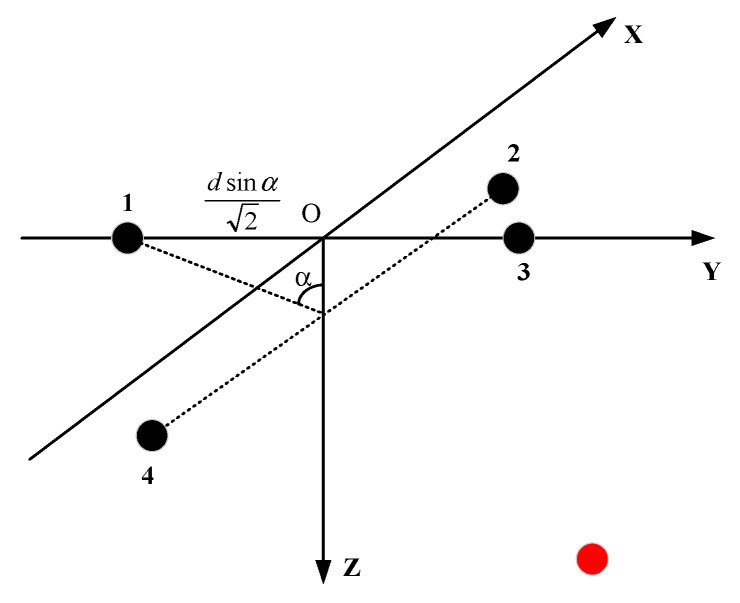
The four-element stereo array 1.

**Figure 3 sensors-21-07722-f003:**
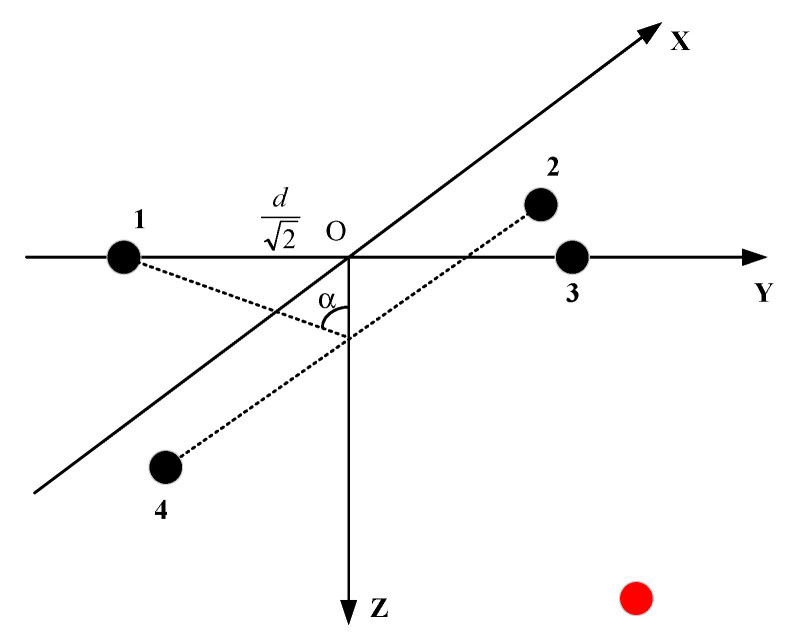
The four-element stereo array 2.

**Figure 4 sensors-21-07722-f004:**
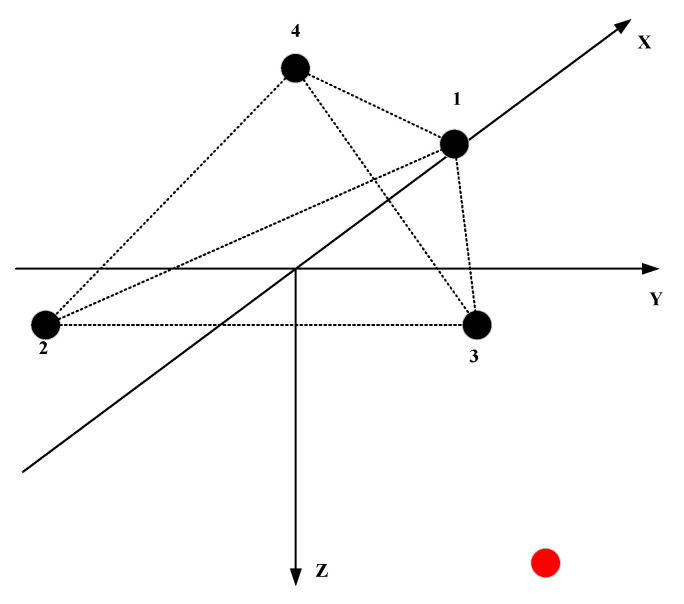
Four-element regular triangular pyramid array.

**Figure 5 sensors-21-07722-f005:**
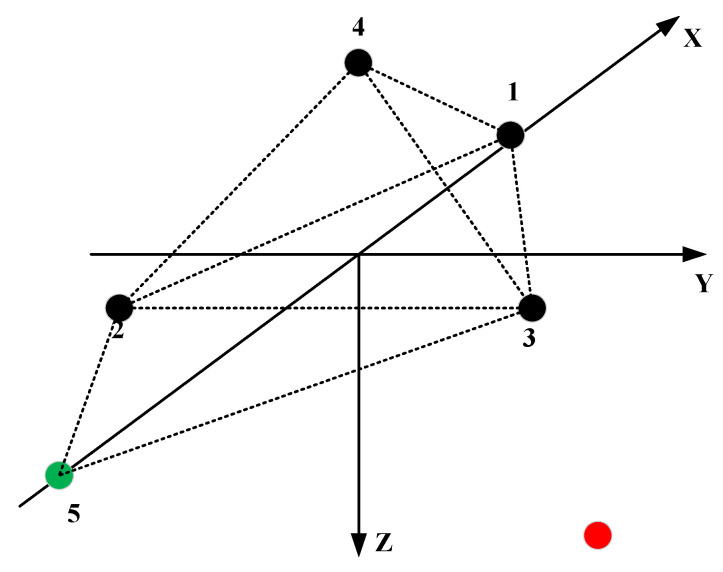
The virtual planar array.

**Figure 6 sensors-21-07722-f006:**
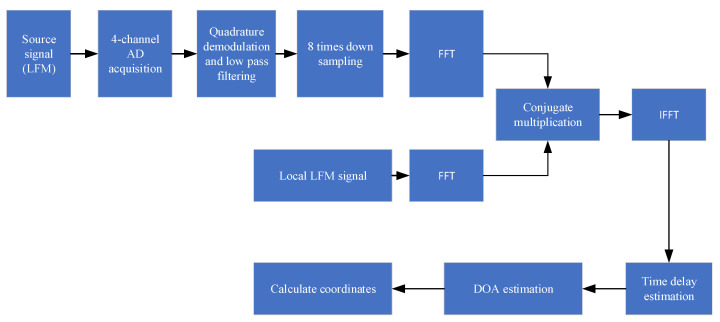
Algorithm processing flow.

**Figure 7 sensors-21-07722-f007:**
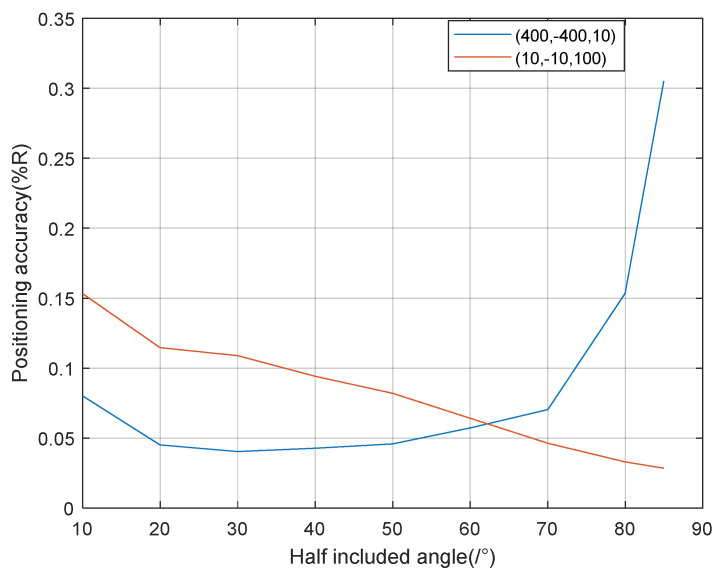
Positioning accuracy of stereo array 1 at different half included angles.

**Figure 8 sensors-21-07722-f008:**
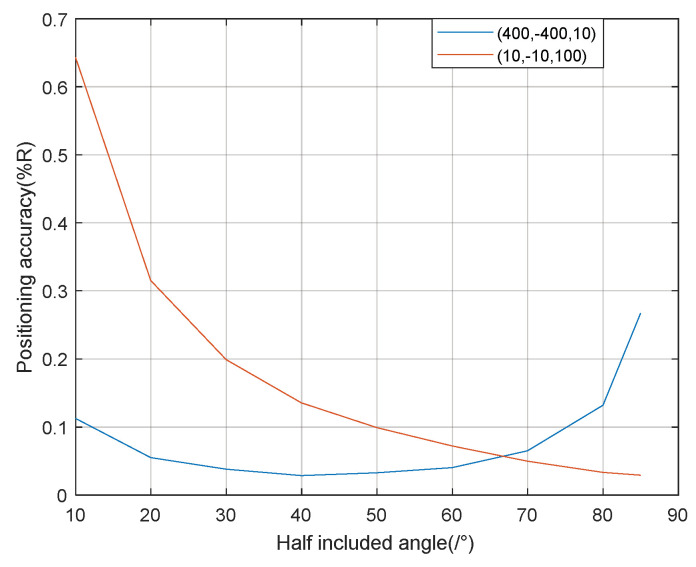
Positioning accuracy of stereo array 2 at different half-included angle.

**Figure 9 sensors-21-07722-f009:**
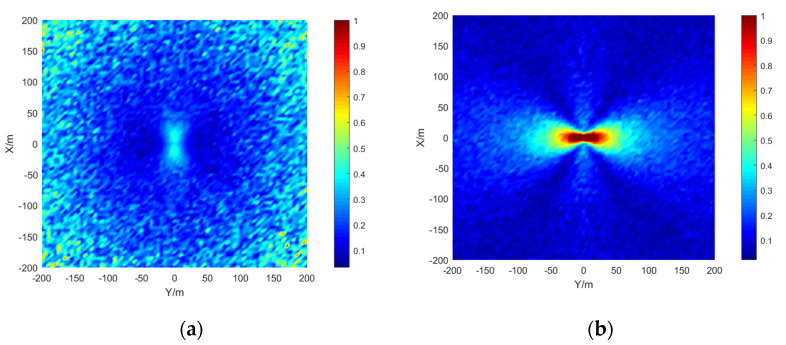
Positioning accuracy of stereo array 1. (**a**) parallel wave algorithm. (**b**) vector projection algorithm.

**Figure 10 sensors-21-07722-f010:**
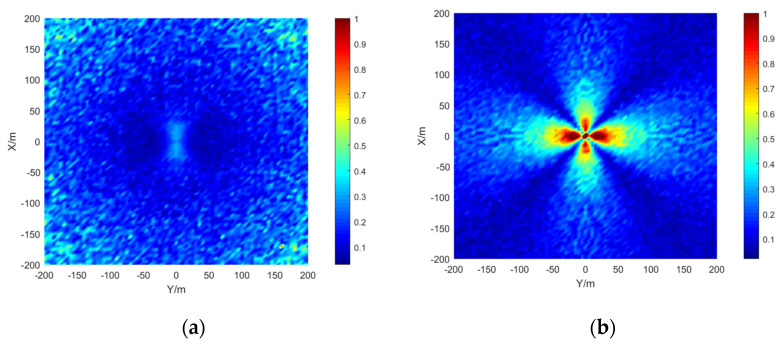
Positioning accuracy of stereo array 2. (**a**) parallel wave algorithm. (**b**) vector projection algorithm.

**Figure 11 sensors-21-07722-f011:**
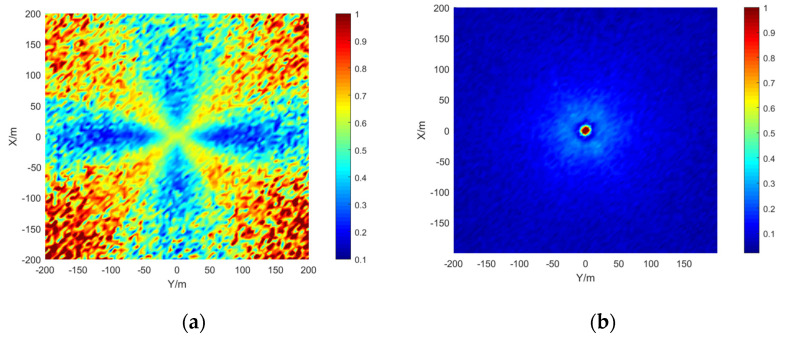
Positioning accuracy of stereo array 3. (**a**) parallel wave algorithm. (**b**) vector projection algorithm.

**Figure 12 sensors-21-07722-f012:**
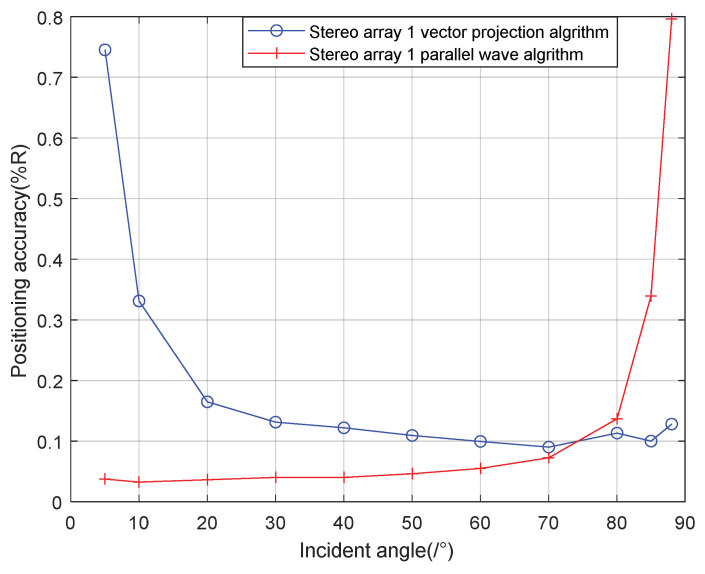
Positioning accuracy of stereo array 1 at different incident angles.

**Figure 13 sensors-21-07722-f013:**
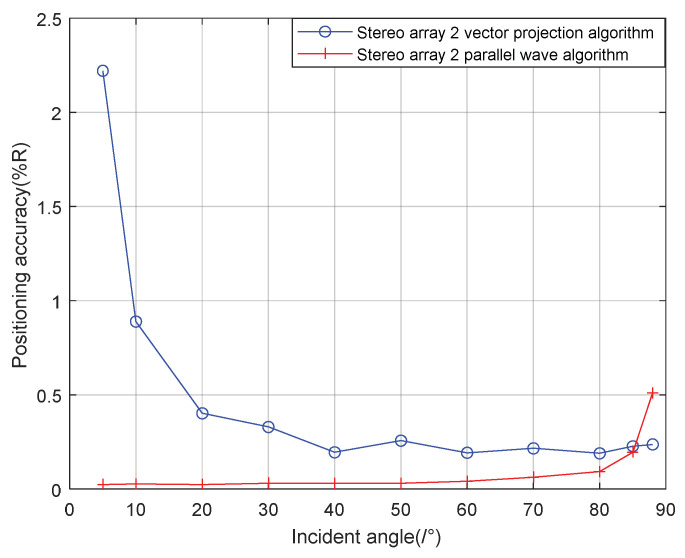
Positioning accuracy of stereo array 2 at different half incident angles.

**Figure 14 sensors-21-07722-f014:**
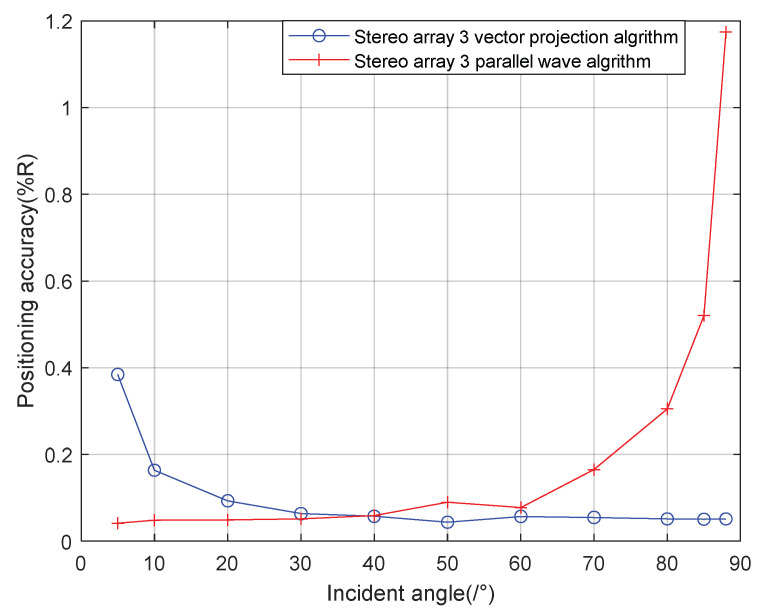
Positioning accuracy of stereo array 3 at different half included angles.

**Figure 15 sensors-21-07722-f015:**
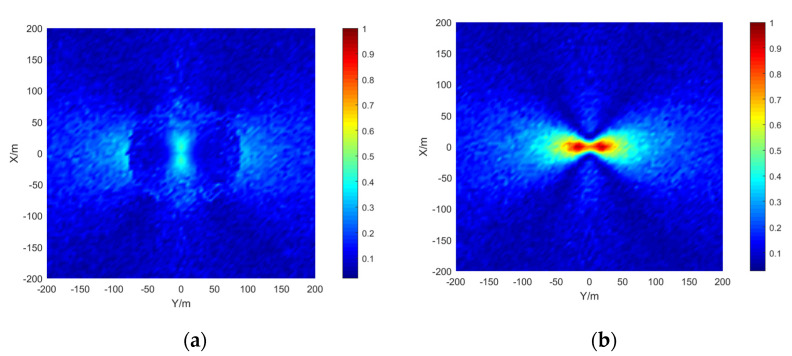
Positioning accuracy of stereo array 1. (**a**) combination algorithm. (**b**) literature 18 algorithm.

**Figure 16 sensors-21-07722-f016:**
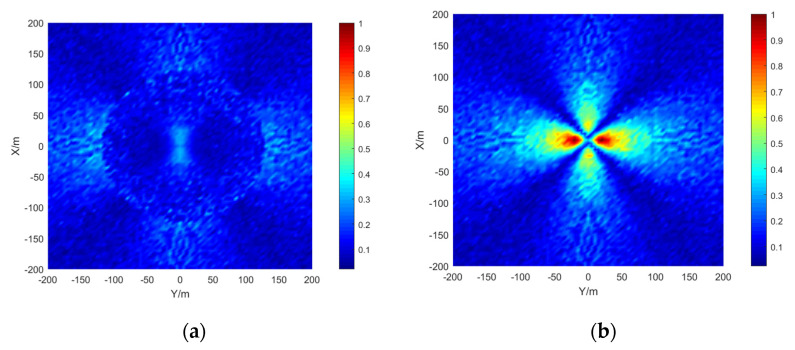
Positioning accuracy of stereo array 2. (**a**) combination algorithm. (**b**) literature 18 algorithm.

**Figure 17 sensors-21-07722-f017:**
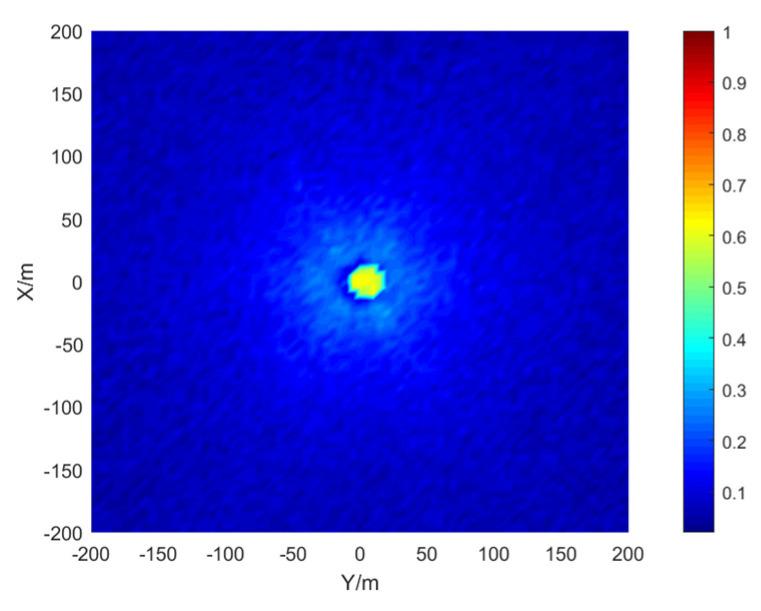
Positioning accuracy of stereo array 3 under combination algorithm.

**Figure 18 sensors-21-07722-f018:**
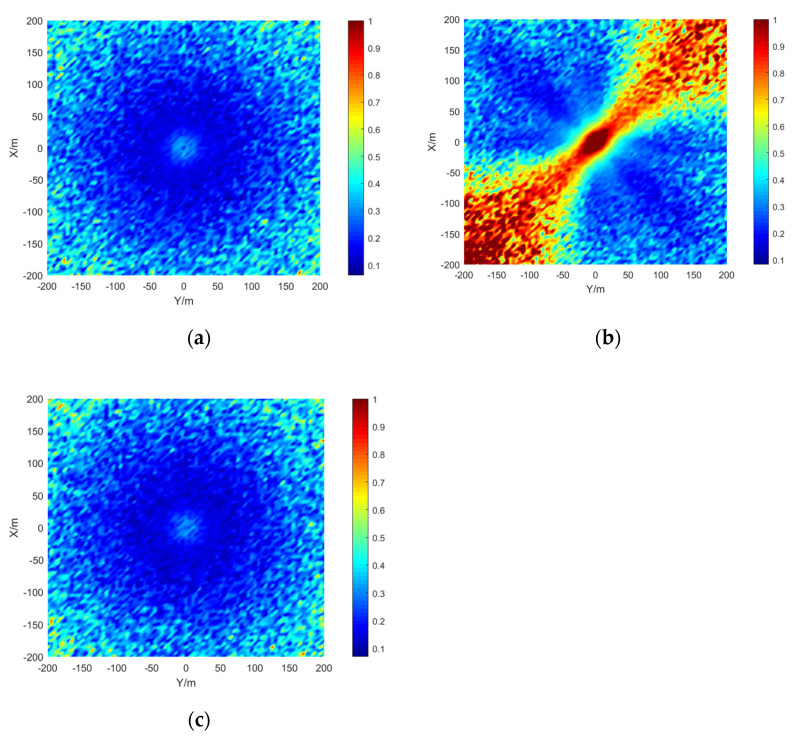
Positioning accuracy of planar array. (**a**) the orthogonal 8-element array. (**b**) the non-equidistant quaternary array. (**c**) the orthogonal quaternary array.

**Figure 19 sensors-21-07722-f019:**
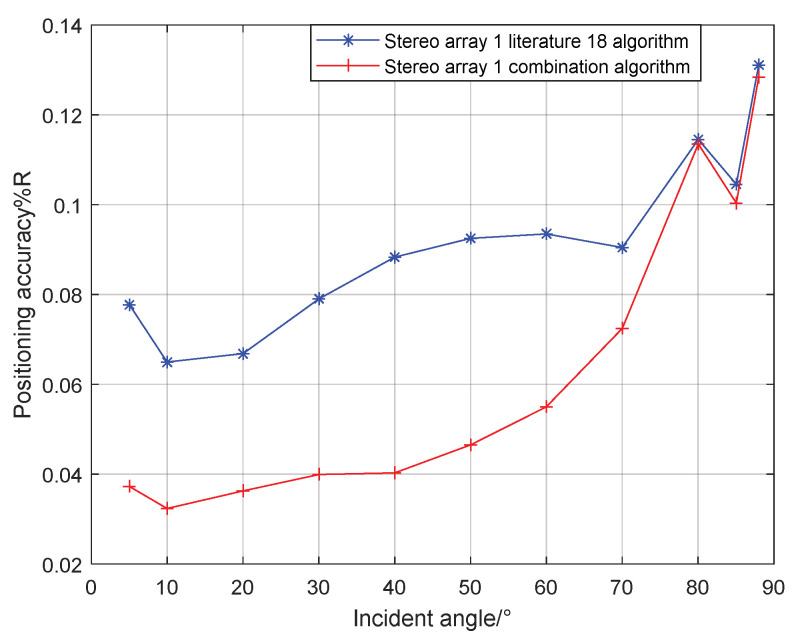
Comparison between combination algorithm and literature 18 algorithm for stereo array 1.

**Figure 20 sensors-21-07722-f020:**
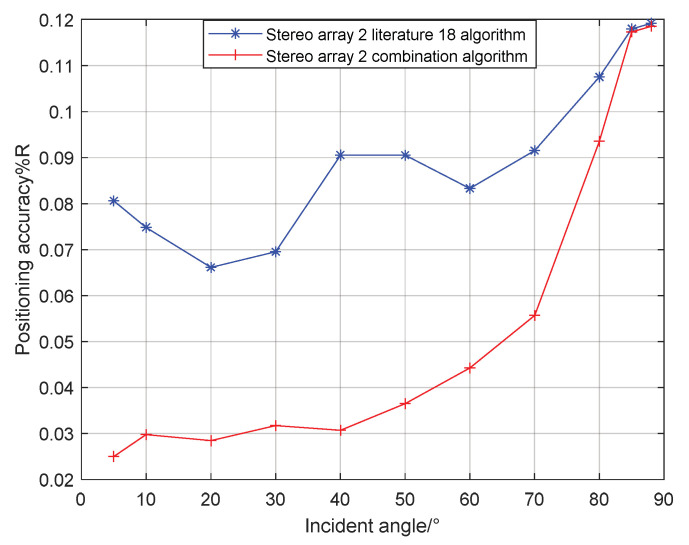
Comparison between combination algorithm and literature 18 algorithm for stereo array 2.

**Figure 21 sensors-21-07722-f021:**
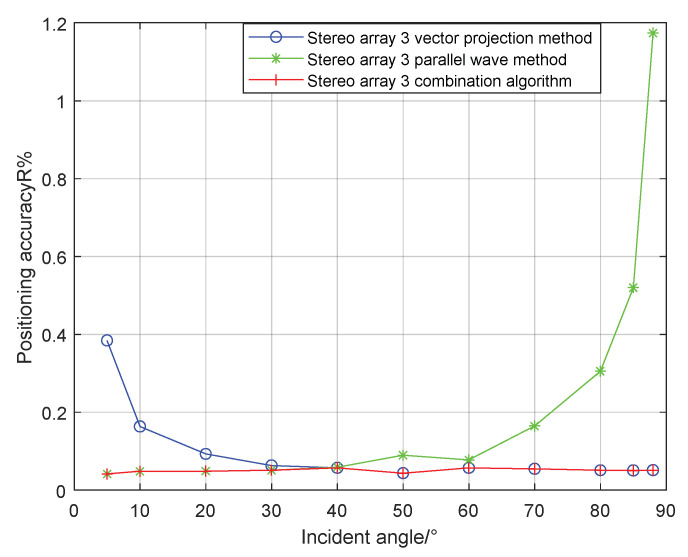
Comparison between combination algorithm and precombination algorithm for stereo array 3.

**Figure 22 sensors-21-07722-f022:**
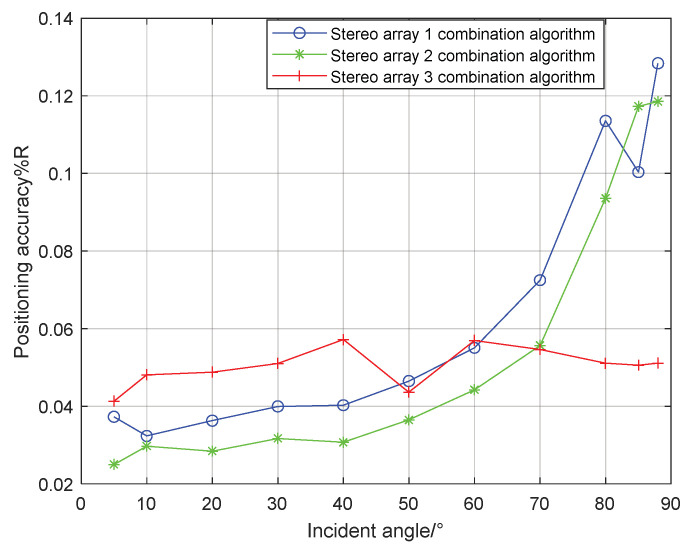
Comparison of three stereo arrays under combination algorithm.

**Table 1 sensors-21-07722-t001:** The localization processing time.

Array	Stereo Array 1	Stereo Array 2	Stereo Array 3	Literature 18	Orthogonal 8-Element Array	Non-Equidistant Quaternary Array	Orthogonal Quaternary Array
Localization time(s)	0.0939	0.0880	0.0946	0.0937	0.0970	0.0520	0.0628

## Data Availability

Not applicable.

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
