# Peer review of "Positioning Combination Method of USBL Using Four-Element Stereo Array"

_sensors, 2021, doi:10.3390/s21227722_

Round 1

Reviewer 1 Report

For higher positioning accuracy, the manuscript proposed two kinds of four-element stereo arrays. It is interesting. However, there are a few issues.

1)There are some format mistakes(e.g. Section 5). Please correct them.

2)The efficiency of the proposed method is not considered. Please make comparisons with that of other related methods.

3)There are numerous kinds of element array, e.g., the non-equidistant quaternary array, the orthogonal eight elements array and orthogonal quaternary array. Checking to make a performance comprehensive comparison with them to show the innovation.

Reviewer 2 Report

The authors presented an ultra-short baseline (USBL) combined location method grounded on three four-component stereo matrices. This paper analyzes the positioning performance of three matrices. Combined with the traditional cross-planar array positioning method, a set of positioning strategies to change the two positioning methods under different incident angles is designed. Simulation results demonstrated that the analyzed matrices could locate stably at different incident angles and improve the overall positioning performance of the matrix.

Strengths

A strong point is an interesting topic and a good application of mathematical knowledge.

Weakness

The weakness of this interesting paper is its modest conclusions. From such extensive material (over 16 pages of text and over 20 figures), the authors drew only 9 lines of conclusions. There are also no conclusions for further research.

Noticed mistake/errors

Line 62-66 are redundant in my opinion, as they add nothing new. Instead, it would be better to briefly summarize the analysis of the state of the issue and present some conclusions on this.

Small errors:

Line 37: What is cuie? If it is (first) name of the author should be Cuie.

Figure 7, Figure 8, Figure 12, Figure 13, Figure 14:

The titles above the chart are redundant as the same information are in the figure subtitle.

Markers on the axis are redundant. Without them, the chart will certainly be easier to read.

Figure 9, Figure 10, Figure 11:

The titles above the chart are redundant. The same pieces of information are in the figure subtitles.

In chapter 4 order number of figures are failed because the figure numbers are not continued from the previous chapter. For all Figures, I have identical objections as for previous figures.

Round 2

Reviewer 1 Report

The author made comprehensive revisions, however, there are a few issues:

1)There are many grammar errors. Please polish it in professional way.

2) There still exists some format mistakes, please check the whole paper including the reference part.

3) In this manuscript, the punctuation marks are misused, e.g. the some brackets are in chinese. Check the manuscript and correct the mistakes.
